# Effects of Lard and Vegetable Oils Supplementation Quality and Concentration on Laying Performance, Egg Quality and Liver Antioxidant Genes Expression in Hy-Line Brown

**DOI:** 10.3390/ani11030769

**Published:** 2021-03-10

**Authors:** Junnan Zhang, Jiajing Chen, Jing Yang, Sijia Gong, Jiangxia Zheng, Guiyun Xu

**Affiliations:** Key Laboratory of Animal Genetics and Breeding of the Ministry of Agriculture, National Engineering Laboratory for Animal Breeding, Department of Animal Genetics and Breeding, College of Animal Science and Technology, China Agricultural University, Beijing 100193, China; cauzhangjn@163.com (J.Z.); kevin2019.3.11@gmail.com (J.C.); yyyangjing2021@163.com (J.Y.); g1965176512@163.com (S.G.); jxzheng@cau.edu.cn (J.Z.)

**Keywords:** oil, laying hens, antioxidant genes, egg quality, production performance

## Abstract

**Simple Summary:**

Adding oils into feeds is essential to the growth and production performance of laying hens. As the main economic benefits of laying hens come from eggs; the quality assurance of eggs is crucial for producers. The term egg quality contains many indicators, including egg shape index, egg weight, yolk weight, yolk color, albumen height, and haugh unit, which is an important index to measure the freshness of eggs. While the oils will oxidize during storage, and feeding with oxidized oil will affect the egg quality and nutritional value. Herein, the Hy-line brown laying hens were fed diets with different types, concentrations, and quality (normal or oxidized) of oil. The results showed that dietary oils quality significantly affect the egg qualities and the expression of liver antioxidant genes, providing useful information for laying hens.

**Abstract:**

This study examined the effects of various types, quality, and levels of dietary oils on laying performance and the expression patterns of antioxidant-related genes in Hy-line brown laying hens. A total of 720 40-week-old Hy-line brown laying hens were fed the same corn-soybean basal meals but containing 0.5 or 1.5% normal or oxidized soybean oil or lard, a total of 8 treatments. The results showed that laying rate (LR) and fatty acids of raw yolk were significantly correlated dietary type of oil (*p* < 0.05). With the increasing concentration of normal oil, it significantly increased LR and decreased feed conversion ratio (FCR, feed/egg) and albumen height of laying hens. The oxidized oil significant decreased the production performance of laying hens; and adding 1.5% of oxidized lard into feeds could destroy the integrity of yolk spheres of cooked yolk. mRNA expression of liver antioxidant-related genes increased when dietary oxidized oils were added into feeds. By comparing different qualities oil effect on antioxidant-related genes, the expression of Glutathione S-Transferase Theta 1 (*GSTT1*), Glutathione S-Transferase Alpha 3 (*GSTA3*), Glutathione S-Transferase Omega 2 (*GSTO2*), and Superoxide Dismutase 2 (*SOD2*) were increased when dietary oils were oxidized, in which change of the *GSTO2* expression was the most with 1.5% of oxidized soybean oil. In conclusion, the ideal type of oil for Hy-line brown layer hens is soybean comparing with lard in a corn-soybean diet, avoiding using of oxidized oil.

## 1. Introduction

Oil is an important supplementation for animal feed. Oil plays numerous roles, such as improving palatability, feed intake, and the energy in feed, enhancing the immunity of body and decreasing the frequency of disease [1,2]. Subsequently, it was widely used in livestock and poultry feed and there is increasing concern the addition of oil in feed from poultry production. Currently, the effect of oil on production performance had been reported many times. The average daily gain (ADI) increased 10.8% and feed conversion ratio (FCR) decreased 11.5% with adding 2% peanut oil in the feed for broiler chickens [3]. Moreover, adding 6% canola oil could significantly decrease the laying rate (LR), average egg weight (AEW) and average daily feed intake (ADFI) [4].

During the storage of oils, the unsaturated fatty acids (UFA) are easily affected by light, heat, oxygen, and microorganisms [5], causing oxidation reactions to generate peroxide, malondialdehyde (MDA), and other oxides, which reduce the quality of oils and animal products [6]. The oxidation of oils in feed is one of the most common factors to cause the oxidative stress in animals [7]. Oxidative stress is a phenomenon caused by an imbalance between production and accumulation of reactive oxygen species (ROS) in cells and tissues and the ability of a biological system to detoxify these reactive products [8]. Therefore, the animal ingests oxidized oils, the peroxide, malondialdehyde (MDA), and other oxides enter the digestive tract and are absorbed into blood to damage the cellular structure and affect various physiological functions [9]. In the process, the body creates mechanisms to respond to cellular oxidative stress, in which superoxide dismutase 2 (*SOD2*) converts the superoxide anion (O^2−^) in ROS to hydrogen peroxide (H_2_O_2_) [10]; under the action of catalase, H_2_O_2_ is oxidized into H_2_O and O^2^, which are finally utilized and cleared through water metabolism and respiration of the body [11]. Another way is to use glutathione-s-transferase (GST), thionredoxin, Vitamin E, and other reducing substances to bind to ROS and block the chain reaction, so as to maintain cell stability [12]. Of these, Glutathione S-Transferase Theta 1 (*GSTT1*), Glutathione S-Transferase Alpha 3 (*GSTA3*), and Glutathione S-Transferase Omega 2 (*GSTO2*) were distributed in cytoplasm, mitochondria, and cell membranes [13], covalently binding to glutathione to metabolize the intermediate products of oxidative stress [12]. Therefore, we speculated that the antioxidant gene expression level in laying hens significantly increased, indicating the occurrence of oxidative stress.

So far, oil that is the most commonly used energy feed is favored by producers [14], as it plays an important role in the production performance and egg quality of laying hens, and little work has been done to integrally assess the effect of oil on egg quality, production performance and expression of antioxidant genes of laying hens in details. The objective of the present study was to investigate the production performance and correlation of liver *GSTT1*, *GSTA3*, *GSTO2* and *SOD2* expression and explore the differences of egg quality with the lard and vegetable oil supplementation of different qualities and concentrations for Hy-line Brown laying hens.

## 2. Materials and Methods

### 2.1. Experimental Design and Diets

A total of 720 Hy-line brown layers of 40 weeks based upon similar body weight of 2000.79 ± 166.18 g/bird were selected from breed base of China Agriculture University (Zhuozhou, China) and randomly assigned to 8 dietary treatments, 3 replicates per treatment, 3 replicates with 90 hens. All the chickens housed in cages as say three per cage of 1350 cm^2^ separately under a light–dark cycle of 16 h light and 8 h dark (16L:8D), equipped with an individual feeder and water. Birds were allowed to adapt to experimental diets in the layer house for 1 week; the entire experiment lasted for 3 weeks. According to the assigned treatment groups, layers were fed with corn-soybean meals containing either 0.5 or 1.5% of normal or oxidized lard or soybean oil, the corn-soybean feed was stored in a dark, 4 °C, and was produced within one month place. Ingredients and nutrient composition of the experimental diets are shown in Table 1. Performance of each laying hen was monitored and eggs were collected on a daily base. Body weight gain (BWG), FCR (feed/egg), ADFI, AEW, and LR were calculated every week.

### 2.2. The Production of Oxidized Oil and Measurement of Oil

The normal oil was exposed to the air and allowed to oxidize until lard became a turbid, gray, and tainted liquid and vegetable oil became turbid, deep yellow, and tainted liquid. Subsequently, the normal and oxidized soybean oil and lard of our experiment were determined by the Chinese food safety standard—Vegetable oil (GB 2716-2018), and Chinese food safety standard—Lard (GB/T 8927-2006). The main indicators include acid value (mg/g), peroxide value (g/100g), and MDA (mg/kg).

### 2.3. Test of Egg Quality

The egg shape index (ESI) was slightly modified from these measurements according to previous research, using the formula: ESI = egg length/egg width [15,16]. Eggshell strength (ESS) was detected by the eggshell strength tester Model-II (Robotmation, Tokyo, Japan). The egg weight (EW), yolk weight (YW), albumen height (AH), yolk color (YC), and haugh unit (HU) were obtained using an EMT5200 multi-function egg tester (Robotmation, Tokyo, Japan). The HU was calculated as 100 log(H + 7.57 − 1.7 W^0.37^) where H = thick albumen height (mm), and W = egg weight (g) [17,18]. Furthermore, the eggshell weight (ESW) was obtained with an electronic balance (YP601N, Qinghai co., ltd, Shanghai, China) after drying and removing the membrane of the eggshell. We obtained the eggshell thickness (EST) in three zones (at both ends and the equator), taking the average of the three measurements. The yolk percentage (YP) was acquired by the formulas YP = YW/EW [19]. Finally, we separated the yolk into a 6 cm diameter petri dish, selected the yolk membrane, and adjusted the yolk liquid to the same height. Then the viscosity of yolk was measured by TA-XT plus texture analyzer (Stable Micro Systems, Godalming, UK) with the P50 probe. The parameter was: 2.00 mm/s of pre-pressure, 0.50 mm/s of down-pressure, 10.00 mm/s of up-pressure and the trigger force was 5.0 g.

### 2.4. Determination of Fatty Acids in Raw and Cooked Egg Yolk

At the end of the experiment, six eggs were randomly selected from each treatment group, in which three raw yolk were directly separated from eggs and mixed evenly. The other three eggs were placed in boiling water for 15 min, removed and cooled to room temperature, peeled and mixed evenly. The fatty acid content of raw and cooked yolk was determined according to Chinese food safety standard (GB 5009.168-2016).

### 2.5. The Sample Preparation for Scanning Electron Microscopy (SEM) Observations

A total of three eggs were randomly selected from each treatment group according the preceding description to make cooked yolk. The cooked egg yolks were sliced with a knife into pieces of about to 5 × 5 × 3 mm^3^ at the center of yolk. In order to prevent sample disintegration during the experimental process, specimens were fixed in 2.5% glutaraldehyde (PH7.2-7.4) for 4–6 h. Each specimen was then washed with phosphate-buffered saline (PBS) 4 times, 20 min each. The samples were then dehydrated using (30%, twice; 50%, twice, 70%, once; 90%, once and 100%, once), graded ethanol. The specimens were transferred into iso-amyl acetate solution for 3 times, 20 min each and were dried [20]. Finally, the samples were coated by sputtering with IB-3 ion sputtering (Hitachi, Tokyo, Japan) [21].

A Hitachi SU8010 SEM (Hitachi High-Technologies, Tokyo, Japan) was used in this study with instructions. For each sample observed by SEM, micrographs at different magnifications (100×, 300×, 700×) were recorded. Each magnification that had 3 micrographs was analyzed, indicating each treatment with 9 micrographs.

### 2.6. Tissue Sampling and Preparation

At the end of the trial, we randomly selected a chicken from each replicate and killed the chickens by cervical dislocation, a total of 24 chickens. The liver tissues were collected and were immediately frozen in liquid nitrogen for later determination of mRNA expression. The whole procedure for collecting the eggs and livers samples was carried out in strict accordance with the protocol approved by the Animal Welfare Committee of China Agricultural University (Permit number: DK996).

### 2.7. Total RNA Extraction, Reverse Transcription, and Real-Time Quantitative PCR

Total RNA was extracted from the liver tissue using Trizol reagent (Invitrogen, Carlsbad, CA, USA) according to the manufacturer’s instructions. Concentrations of RNA were measured by absorbance at 260 nm with Nanodrop 2000 spectrophotometer (Thermo Scientific, Waltham, MA, USA). Reverse transcription of total RNA to cDNA was conducted with PrimerScriptH RT reagent Kit (TaKaRa, Shiga, Japan).

Gene expression of liver *GSTT1*, *GSTA3*, *GSTO2* and *SOD2* were determined by real-time quantitative PCR (qPCR) with β-actin and *GAPDH* as the internal standard. The primers used and information of PCR products are listed in Appendix A. qRT-PCR was performed in a final reaction volume of 20 μL in an iCycler iQ 5 Multicolor Real-Time PCR Detection System (Bio-Rad, Foster, CA, USA). The reaction mixture contained 1 μL cDNA, 9 μL 2.5×RealMasterMix/20×SYBR solution (Tiangen, Beijing, China), 1.6 μL primers and ddH2O up to 20 μL. The following protocol was used: 95 °C for 10 min; 40 cycles of 95 °C for 10 s, 60 °C for 10 s, and 72 °C for 10 s. Each sample was amplified in triplicate and amplification efficiency was determined by standard curves to ensure equal efficiency between target genes and the internal control standard. The 2^−ΔΔCt^ method was used to calculate the expression of the target gene, as previously described [22].

### 2.8. Statistical Analysis

For all statistical analyses, each replicate served as the experimental unit. All data were analyzed using the Statistical Package for the Social Sciences (SPSS) 25 to conduct one-way ANOVA and Duncan’s multiple range test, which is a post hoc test to measure specific differences between pairs of means [23]. *p* < 0.05 was considered statistically significant.

## 3. Results

### 3.1. The Indicators of Oil

The appearance of normal soybean oil and was clear and yellow liquid, and normal lard was pure and white solid. However, the oxidized oils showed darker color and turbid liquid and produced odorous smell. After detection, the acid value of normal soybean oil was a little higher than lard, while the acid value of oxidized lard was 8 times that of the normal lard. The MDA level in oxidized lard reached 3.160 (mg/kg). Finally, the content of UFA in the oxidized oil decreased, while the content of saturated fatty acid (SFA) increased (Table 2).

### 3.2. Performance of Laying Hens

The results of the performance of the laying hens are provided in Table 3. Whether it was lard or soybean oil, the ADFI for treatment with oxidized oils showed a downward trend, and the ADFI of HOV group was significantly lower than HNV treatments (*p* < 0.05). The FCR of laying hens fed with normal lard was significantly lower than the normal soybean group (*p* < 0.05). With the oxidized oil used in feeds, the FCR had no differences between high-dose lard and soybean oil groups. AEW and LR with high-dose oxidized oil significantly decreased (*p* < 0.05); there were no differences on AEW with the same quality oil in feeds. Moreover, the BWG was significantly influenced by the quality and type of oil (*p* < 0.05), in which the lard could significantly increase the BWG than the soybean group.

### 3.3. Egg Quality

The results of the egg quality of the laying hens are provided in Table 4. No difference was observed in yolk color (YC), eggshell weight (ESW), eggshell strength (ESS), egg shape index (ESI), and yolk percentage (YP). However, the AH and HU were significantly influenced by the concentration of supplementary (*p* < 0.05), in which AH of HNV and HNL treatments were significantly lower than the LNV and LNL treatments (*p* < 0.05). The AH of oxidized treatments was commonly higher than the normal treatments except HOL group. Similarly to the AH, the HU of laying hens fed with LOL was significantly higher than laying hens fed with LNL (*p* < 0.05).

### 3.4. The Fatty Acid Composition of Raw and Cooked Yolk

In the raw yolk, the top three fatty acids were C18:1n9c (oleic acid), C16:0 (palmitic acid) and C18:2n6c (linoleic acid), in which the C16:0 that was significantly influenced by the concentrations, types and qualities of oils was the most abundant of SFA (*p* < 0.05), while the C17:0 was the least substance. Moreover, the oxidized oils could increase the content of the C18:1n9c, which was the highest UFA and was greatly affected by the types of oil (*p* < 0.05), whereas the least substance was c18:3n6. The content of UFA in the raw yolk was significantly higher than that of SFA (*p* < 0.05), and the oxidized oil significantly increased the overall content of UFA and SFA (*p* < 0.05). Therefore, the total fatty acid (TFA) of raw yolk of laying hens fed with oxidized oil was higher than the lay hens fed with normal oil (Table 5).

In the cooked yolk, the top 3 fatty acids were also C18:1n9c (oleic acid), C16:0 (palmitic acid) and C18:2n6c (linoleic acid), same as the raw yolk. We found that there were irregular effects on fatty acids with various types, qualities, and concentrations of oils in feeds. In the UFA and SFA, C17:0 (margaric acid), C18:1n9c (oleic acid), C18:2n6c (linoleic acid), C20:1 (eicosenoic acid), C20:3n6 (eicosatrienoic acid), and C20:4n6 (methyl arachidonate) were not influenced by the quality, type, and concentration of oil. The content of UFA and SFA decreased significantly fed with HOL, which was the most significant comparing with the other seven treatments (Table 6).

In addition, we found an increase in UFA and SFA in cooked egg by integrated analyzing fatty acid content of raw and cooked yolk, but an opposite consequence was detected under the HOL treatment, the TFA was significantly decreased from 161.48 to 94.45 mg/g.

### 3.5. Microstructure of Cooked Yolk

To further investigate the effect of oil quality in feed on the egg quality, we observed the microstructure of cooked yolk by scanning electron microscopy. The cooked yolk spheres showed the polyhedron shape with a diameter at approximately 40–100 μm and were packed closely together, and the surface and edge of the spheres were uneven and angular. From Figure 1 we could find the yolk spheres were not influenced by the quality in the LNV vs. LOV, HNV vs. HOV, and LNL vs. LOL. However, in the HOL group, there had been a significant change that were the obvious crosslinking among the different yolk spheres and signs of fragmentation in the yolk spheres.

### 3.6. Gene Expression of GSTT1, GSTA3, GSTO2, and SOD2

The gene expression of *GSTT1*, *GSTA3*, *GSTO2,* and *SOD2* were shown in Figure 2. Dietary quality significantly affected the expression of genes encoding antioxidant enzymes in the liver of laying hens (*p* < 0.05). For the low-dose vegetable oils, the expression of *GSTT1* with oxidized oil in feed increased significantly as oil oxidized, the expression of other three genes had no significant differences. While the expression *GSTA3*, *GSTO2* and *SOD2* of high-dose vegetable oils were significantly positively correlated with the degree of oxidation. When the quality of oil of low-dose lard groups had changed, *GSTT1*, *GSTO2* and *SOD2* genes significantly increased, only the expression of *GSTA3* had changed significantly fed with HOL.

## 4. Discussion

The oxidized soybean oil and lard were murky and odorous liquid, indicating that insoluble substances and aldehydes were produced during the oxidation process [24]. The content of MDA that was as one of the final products of oil oxidation could illustrate the degree of oil oxidization [25]. The MDA content in oxidized lard increased by 6 times, however, the increase in oxidized soybean oil was smaller, possibly because the soybean oil contains antioxidant substances, which reduces the oxidation degree [26]. In addition, we could found that our treatments not isoenergetic or isoproteics from Table 1, some effects could be the result of this in the subsequent analysis.

Comparing with all the normal oil groups, the ADFI and AEW both decreased in the oxidized oil groups. The explanation for this might be that oxidized oil decreased feed digestibility, damaged intestinal epithelial cells, affected feed absorption and usage, and decreased production efficiency [27,28]. The strong smell and taste of oxidized oil may also have contributed to the decrease the feed intake. In addition, with the same quality and concentration of oil, soybean oil as a feed additive can improve the laying rate of laying hens better than lard in our study. We suggested add soybean oil to the feed to improve the performance of laying hens, and the oil should be sealed and placed in a cool place to prevent oxidation. For the effect of different oil qualities on egg quality, the HU that reflects the protein content and freshness of the egg had increased significantly in HOV group than that in HNV group. The reason is that the ROS of oxidized oil could promotes the synthesis and secretion of antioxidant enzymes in the liver [29], and enhances the secretion capacity of the secretory cells in the magnum, which secreted the egg white to encase the yolk [30]. Therefore, when we evaluate the quality of eggs, we are not inclined to take HU as a single standard for evaluation.

Dietary oil could affect the lipid metabolism of laying hens to alter the lipid composition of yolk [4]. Rowghani et al. reported that a significant increase in cholesterol of yolk was observed in 24 weeks Hy laying hens when fed 3 or 5% level of canola oil [31]. The SFA content of raw yolk decreased and the UFA increased as the concentration of the same quality of oil increases in our study. The fatty acid of raw yolk in the oxidized oil group is higher than in the normal oil group may be due to the lower LR, which increases the amount of fatty acid deposited. In addition, the α-linolenic acid (C18:3n3), eicosadienoic acid (C20:2), and docosahexaenoic acid (C22:6n3) of cooked yolk in soybean group is higher than in lard group. With the increasing concentration, the content of linoleic acid (C18:2n6c), palmitoleic acid (C16:1), and UFA had increased, which was consistent with the variation of fatty acid in raw yolk. The less content of fatty acid could was observed obviously in cooked yolk rather than in raw yolk, the reason is that yolk produced lots of volatile substances while the eggs were being cooked to reduce various of fatty acids [32]. Finally, the destruction of the integrity of yolk spheres were observed in the HOL group under the SEM.

Different dietary quality significantly affected antioxidant-related genes expression of livers in laying hens during the 3-weeks experimental period. From Figure 1, we could found that the *GSTT1*, *GSTA3*, *GSTO2* and *SOD2* genes were significantly increased in different degrees as the oils oxidized. These data suggest a significantly positive feedback effect of the oxidation of oils on the antioxidant gene of liver. *GSTT1*, *GSTA3* and *GSTO2* genes belongs to glutathione transferase (GST) family [33], which are widely distributed in various cells and catalyze the endogenous and exogenous detrimental electrophiles to conjugate them with reduced glutathione [34], making the oxidized substances more easily pass through the cell membrane, thus being excluded from the body [35]. The discovery of *SOD2*, which specifically catalyze the dismutation of superoxide radicals (O^2−)^ to hydrogen peroxide (H_2_O_2_) and oxygen. The H_2_O_2_ can react with ferrous ion to form hydroxyl [36]. Moreover, the *SOD2* gene has ability to remove the ROS that is produced by respiratory chain to maintain the stability of mitochondrial DNA [37].

## 5. Conclusions

In conclusion, under the same quality and concentration of oil, adding soybean oil to the feed of Hy-brown laying hens can improve production performance better than lard. Moreover, the expression level of antioxidant genes was significantly increased after feeding with oxidized oil, we should avoid utilizing oxidized oil to add into the feed to prevent the oxidative stress in vivo. Our findings provided new insights into the effect of different types, concentrations, and qualities of dietary oils on the production performance and egg quality in laying hens.

## Figures and Tables

**Figure 1 animals-11-00769-f001:**
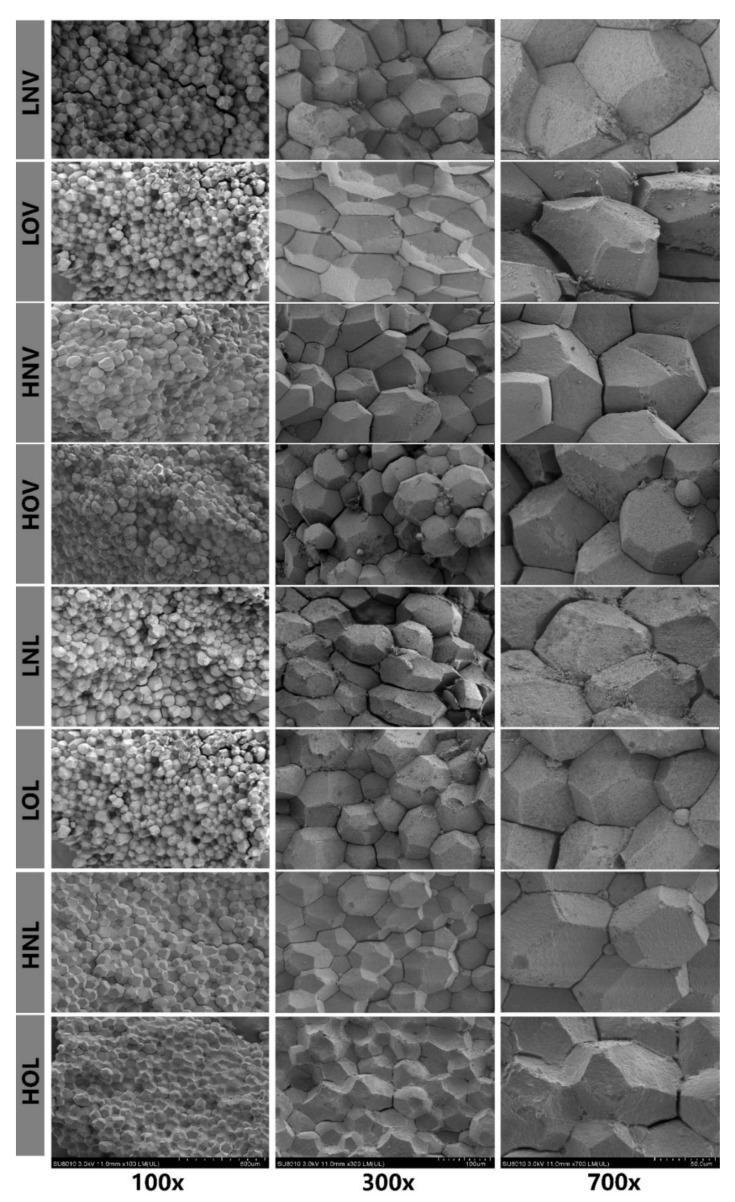
Micrographs of cooked yolk observed by SEM-SU8010 at different magnifications. The abscissa indicates the different magnifications. LNV = low-dose normal vegetable oil, HNV = high-dose normal vegetable oil, LNL = low-dose normal lard, HNL = high-dose normal lard, LOV = low-dose oxidized vegetable oil, HOV = high-dose oxidized vegetable oil, LOL = low-dose oxidized lard, HOL = high-dose oxidized lard.

**Figure 2 animals-11-00769-f002:**
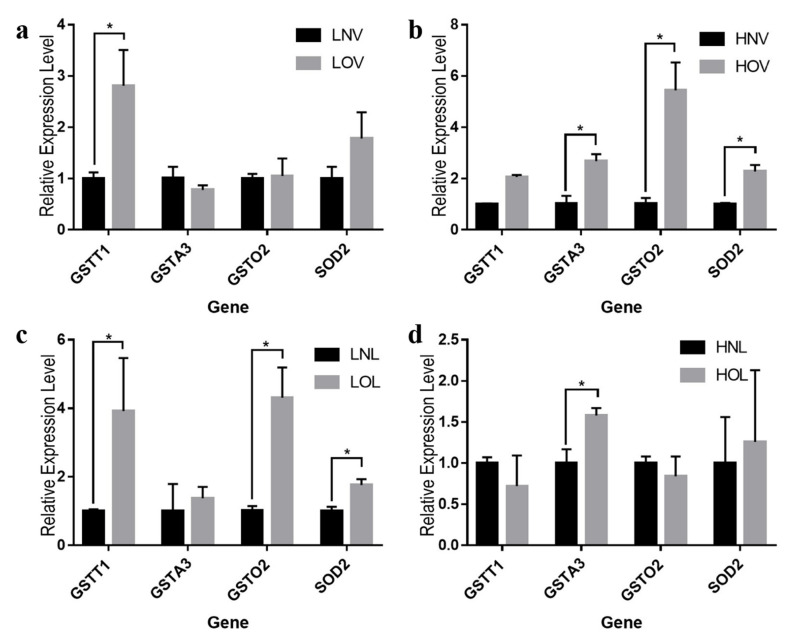
Effects of different dietary qualities on gene expression of *GSTT1*, *GSTA3*, *GSTO2* and *SOD2* of livers. The abscissa indicates the gene name; and the ordinate value is normalized to *β-actin* and *NAPDH*, results are expressed as 2^−^^ΔΔCt^ (*n* = 24). LNV = low-dose normal vegetable oil, HNV = high-dose normal vegetable oil, LNL = low-dose normal lard, HNL = high-dose normal lard, LOV = low-dose oxidized vegetable oil, HOV = high-dose oxidized vegetable oil, LOL = low-dose oxidized lard, HOL = high-dose oxidized lard. * *p* < 0.05.

**Table 1 animals-11-00769-t001:** Composition of laying hens’ diets.

Items ^1^	Groups ^2^							
	LNV	HNV	LNL	HNL	LOV	HOV	LOL	HOL
Ingredient, %								
Corn	63.42	60.45	60.36	60.07	63.42	60.45	63.36	60.07
Soybean meal	24.56	25.13	24.57	25.21	24.56	25.13	24.57	25.21
Normal soybean oil	0.5	1.5	-	-	-	-	-	-
Oxidized soybean oil	-	-	-	-	0.5	1.5	-	-
Normal lard	-	-	0.5	1.5	-	-	-	-
Oxidized lard	-	-	-	-	-	-	0.5	1.5
Stone powder	8.5	8.5	8.5	8.5	8.5	8.5	8.5	8.5
Premix ^3^	3	3	3	3	3	3	3	3
Zeolite powder	0	1.4	0.05	1.7	0	1.4	0.05	1.7
Antioxidant ^4^	0.02	0.02	0.02	0.02	0.02	0.02	0.02	0.02
Total	100	100	100	100	100	100	100	100
Nutrient composition, %	-	-	-	-	-	-	-	-
AME, MJ/kg	11.37	11.38	11.38	11.45	11.37	11.38	11.38	11.45
Protein	17.28	17.30	17.28	16.83	17.28	17.30	17.28	16.83
Calcium	3.14	3.14	3.14	3.06	3.14	3.14	3.14	3.06
Phosphorus	0.33	0.32	0.33	0.32	0.33	0.32	0.33	0.32
Lysine	0.89	0.89	0.89	0.89	0.89	0.86	0.89	0.86
Methionine	0.28	0.28	0.28	0.28	0.28	0.27	0.28	0.27

^1^ AME = apparent metabolizable energy. ^2^ LNV = low-dose normal vegetable oil, HNV = high-dose normal vegetable oil, LNL = low-dose normal lard, HNL = high-dose normal lard, LOV = low-dose oxidized vegetable oil, HOV = high-dose oxidized vegetable oil, LOL = low-dose oxidized lard, HOL = high-dose oxidized lard. ^3^ Premix: Mineral premix provided per kg of diet: Mn, 100 mg; Fe, 80 mg; Zn, 75 mg; Cu, 8 mg; I, 0.35 mg; Se, 0.15 mg. Vitamin premix provided per kg of diet: vitamin A, 12,500 International units (IU); vitamin D3, 2500 IU; vitamin E, 30 IU; vitamin K3, 2.65 mg; vitamin B1, 2 mg; vitamin B2, 6 mg; vitamin B12, 0.025 mg; biotin, 0.0325 mg; folic acid, 1.25 mg; pantothenic acid, 12 mg; niacin, 50 mg. ^4^ Antioxidant = ethoxyquin.

**Table 2 animals-11-00769-t002:** The index of different quality lard and vegetable oil.

Items ^1^	Normal Soybean Oil	Normal Lard	Oxidized Soybean Oil	Oxidized Lard
Appearance	The clear, yellow and scented liquid	The pure, white and scented solid	The murky, yellow and odorous liquid	The murky, grey and odorous liquid
Acid Value, mg/g	1.130	0.758	1.340	5.805
Peroxide Value, g/100 g	0.085	0.060	0.343	0.079
MDA, mg/kg	0.435	0.497	1.890	3.160
SFA, %	47.156	46.887	56.276	55.813
UFA, %	48.347	48.002	34.778	34.539

^1^ MDA =malondialdehyde, SFA = saturated fatty acid, UFA = unsaturated fatty acid.

**Table 3 animals-11-00769-t003:** Effect of vegetable oil and lard with different qualities and concentrations on production performance of laying hens ^1^.

Item ^2^	Groups ^3^							
	LNV	HNV	LNL	HNL	LOV	HOV	LOL	HOL
ADFI, g	115.00 ± 3.77 ^a^	115.47 ± 4.18 ^a^	111.78 ± 4.93 ^ab^	112.93 ± 10.22 ^ab^	107.16 ± 1.19 ^ab^	105.44 ± 4.78 ^b^	102.62 ± 5.10 ^b^	107.07 ± 1.99 ^ab^
FCR, g/g	2.14 ± 0.12 ^a^	2.03 ± 0.11 ^b^	2.14 ± 0.10 ^a^	1.89 ± 0.15 ^c^	1.88 ± 0.13 ^c^	1.93 ± 0.09 ^bc^	2.01 ± 0.11 ^b^	1.96 ± 0.07 ^bc^
AEW, g	63.40 ± 0.69 ^ab^	64.07 ± 0.97 ^a^	62.83 ± 0.53 ^b^	63.04 ± 0.61 ^b^	62.22 ± 2.89 ^bc^	61.89 ± 1.28 ^c^	61.93 ± 0.97 ^c^	61.79 ± 1.40 ^c^
LR, %	87.56 ± 4.03 ^b^	91.77 ± 4.03 ^a^	82.03 ± 4.07 ^c^	92.26 ± 3.79 ^a^	87.47 ± 4.00 ^b^	87.46 ± 4.00 ^b^	82.64 ± 4.10 ^c^	88.89 ± 4.12 ^b^
BWG, g	40.43 ± 222.70	51.64 ± 304.89	35.41 ± 302.74	52.82 ± 221.38	26.99 ± 190.78	31.78 ± 282.30	30.88 ± 225.8	47.52 ± 311.90

^1^ Each value is presented as mean ± standard deviation, ^a–c^ means in a row without a common superscript letter differ (*p* < 0.05), as analyzed by one-way ANOVA. ^2^ ADFI = average daily feed intake, FCR = feed conversion ratio, AEW = average egg weight, LR = laying rate, BWG = body weight gain. ^3^ LNV = low-dose normal vegetable oil, HNV = high-dose normal vegetable oil, LNL = low-dose normal lard, HNL = high-dose normal lard, LOV = low-dose oxidized vegetable oil, HOV = high-dose oxidized vegetable oil, LOL = low-dose oxidized lard, HOL = high-dose oxidized lard; different superscripts within a column indicate significant difference.

**Table 4 animals-11-00769-t004:** Effect of vegetable oil and lard with different qualities and concentrations on egg quality of laying hens ^1^.

Item ^2^	Groups ^3^							
	LNV	HNV	LNL	HNL	LOV	HOV	LOL	HOL
AH, mm	4.95 ± 1.11 ^b^	4.72 ± 1.12 ^c^	5.26 ± 1.04 ^a^	4.95 ± 0.96 ^b^	5.07 ± 1.06 ^ab^	5.13 ± 1.18 ^ab^	5.05 ± 1.08 ^ab^	4.76 ± 1.05 ^bc^
YC	6.36 ± 0.79 ^ab^	6.21 ± 0.75 ^b^	6.14 ± 0.65 ^b^	6.18 ± 0.59 ^b^	6.32 ± 0.72 ^ab^	6.26 ± 0.76 ^b^	6.42 ± 0.60 ^a^	6.38 ± 0.60 ^ab^
HU	64.58 ± 12.71 ^b^	63.66 ± 11.13 ^b^	68.33 ± 10.80 ^a^	66.05 ± 9.53 ^ab^	67.38 ± 9.37 ^ab^	67.89 ± 11.32 ^a^	66.97 ± 10.25 ^b^	65.04 ± 9.71 ^b^
EW, g	6.61 ± 1.09	6.81 ± 0.54	6.80 ± 0.81	6.53 ± 0.67	6.86 ± 0.78	7.19 ± 0.50	6.75 ± 0.73	6.82 ± 0.55
EST, μm	373.16 ± 25.5 ^a^	372.59 ± 32.2 ^a^	368.33 ± 33.08 ^ab^	373.74 ± 25.51 ^a^	368.60 ± 28.83 ^a^	365.98 ± 27.59 ^ab^	352.74 ± 36.96 ^b^	356.20 ± 35.58 ^b^
ESS, *n*/cm^2^	44.34 ± 10.51	45.20 ± 9.81	44.07 ± 10.05	45.34 ± 9.37	43.49 ± 10.78	43.28 ± 10.57	45.25 ± 9.13	43.64 ± 10.21
ESI	1.34 ± 0.05	1.35 ± 0.05	1.35 ±0.19	1.34 ± 0.04	1.35 ± 0.05	1.35 ± 0.05	1.34 ± 0.05	1.34 ± 0.05
YP, %	26.41 ± 1.851	26.4 ± 1.576	26.35 ± 2.113	26.27 ± 3.406	26.09 ± 3.763	25.83 ± 2.615	25.82 ± 3.04	25.7 ± 3.804

^1^ Each value is presented as mean ± standard deviation, ^a–c^ means in a row without a common superscript letter differ (*p* < 0.05), as analyzed by one-way ANOVA. ^2^ AH = albumen height, YC = yolk color, HU = haugh unit, EW = egg weight, EST = eggshell thickness, ESS = eggshell strength, ESI = eggshell index, YP = yolk percentage. ^3^ LNV = low-dose normal vegetable oil, HNV = high-dose normal vegetable oil, LNL = low-dose normal lard, HNL = high-dose normal lard, LOV = low-dose oxidized vegetable oil, HOV = high-dose oxidized vegetable oil, LOL = low-dose oxidized lard, HOL = high-dose oxidized lard; different superscripts within a column indicate significant difference.

**Table 5 animals-11-00769-t005:** Effect of vegetable oil and lard with different qualities and concentrations on fatty acids of raw yolk ^1^.

Item ^2^	Groups ^3^							
	LNV	HNV	LNL	HNL	LOV	HOV	LOL	HOL
C14:0, mg/g	0.43 ± 0.027 ^c^	0.37 ± 0.012 ^d^	0.27 ± 0.007 ^e^	0.44 ± 0.027 ^c^	0.54 ± 0.02 ^ab^	0.56 ± 0.02 ^ab^	0.42 ± 0.029 ^c^	0.6 ± 0.022 ^a^
C16:0, mg/g	34.67 ± 2.348 ^d^	30.49 ± 0.924 ^e^	20.65 ± 0.522 ^f^	31.72 ± 1.442 ^de^	40.09 ± 1.493 ^b^	38.09 ± 1.45 ^c^	32.51 ± 1.659 ^de^	45.87 ± 2.063 ^a^
C16:1, mg/g	4.64 ± 0.305 ^ab^	3.63 ± 0.11 ^b^	2.83 ± 0.071 ^b^	1.94 ± 2.465 ^b^	6.18 ± 0.218 ^a^	5.18 ± 0.218 ^a^	4.33 ± 0.213 ^ab^	5.27 ± 0.225 ^ab^
C17:0, mg/g	0.16 ± 0.013 ^d^	0.19 ± 0.005 ^bc^	0.09 ± 0.002 ^e^	0.2 ± 0.008 ^b^	0.18 ± 0.01 ^c^	0.18 ± 0.01 ^c^	0.17 ± 0.009 ^c^	0.29 ± 0.003 ^a^
C18:0, mg/g	2.9 ± 0.24 ^b^	2.62 ± 0.253 ^b^	1.75 ± 0.496 ^c^	2.69 ± 0.15 ^b^	2.87 ± 0.32 ^b^	2.77 ± 0.31 ^b^	2.58 ± 0.188 ^b^	4.05 ± 0.334 ^a^
C18:1n9c, mg/g	52.18 ± 3.875 ^c^	47.24 ± 0.92 ^c^	32.78 ± 0.885 ^d^	48.41 ± 2.265 ^c^	57.7 ± 2.50 ^b^	52.7 ± 2.4 ^b^	49.99 ± 2.878 ^c^	73.54 ± 3.833 ^a^
C18:2n6c, mg/g	15.95 ± 1.143 ^c^	16.6 ± 0.585 ^b^	8.12 ± 0.214 ^d^	15.42 ± 0.687 ^c^	18.26 ± 0.738 ^b^	18.28 ± 0.75 ^b^	14.8 ± 0.769 ^c^	23.16 ± 1.109 ^a^
C18:3n6, mg/g	0.15 ± 0.008 ^b^	0.16 ± 0.005 ^b^	0.09 ± 0.003 ^c^	0.17 ± 0.012 ^b^	0.21 ± 0.012 ^a^	0.20 ± 0.01 ^a^	0.17 ± 0.008 ^b^	0.21 ± 0.005 ^a^
C20:1, mg/g	0.31 ± 0.029 ^bc^	0.27 ± 0.01 ^c^	0.17 ± 0.005 ^d^	0.27 ± 0.009 ^c^	0.32 ± 0.02 ^b^	0.31 ± 0.02 ^b^	0.28 ± 0.021 ^c^	0.43 ± 0.013 ^a^
C18:3n3, mg/g	4.1 ± 0.3 ^c^	4.5 ± 0.16 ^bc^	1.6 ± 0.02 ^e^	3.3 ± 0.11 ^d^	4.8 ± 0.17 ^b^	4.6 ± 0.17 ^b^	3.0 ± 0.22 ^d^	5.7 ± 0.18 ^a^
C20:2, mg/g	1.7 ± 0.09 ^b^	1.4 ± 0.43 ^b^	0.9 ± 0.01 ^b^	1.7 ± 0.17 ^b^	1.8 ± 0.07 ^b^	1.8 ± 0.06 ^b^	1.7 ± 0.08 ^b^	2.4 ± 0.12 ^a^
C20:3n6, mg/g	0.23 ± 0.016 ^d^	0.27 ± 0.008 ^bc^	0.16 ± 0.004 ^e^	0.21 ± 0.01 ^d^	0.29 ± 0.01 ^b^	0.29 ± 0.02 ^b^	0.26 ± 0.011 ^c^	0.33 ± 0.016 ^a^
C20:4n6, mg/g	2.74 ± 0.201 ^c^	3.11 ± 0.089 ^b^	1.92 ± 0.051 ^d^	2.86 ± 0.13 ^bc^	3.1 ± 0.11 ^b^	3.1 ± 0.12 ^b^	3.19 ± 0.157 ^b^	4.4 ± 0.225 ^a^
C22:6n3, mg/g	0.93 ± 0.071 ^bc^	1.06 ± 0.031 ^b^	0.55 ± 0.015 ^d^	0.88 ± 0.042 ^c^	0.95 ± 0.045 ^bc^	0.96 ± 0.05 ^bc^	0.99 ± 0.126 ^bc^	1.44 ± 0.076 ^a^
SFA, mg/g	38.3 ± 2.165 ^c^	33.8 ± 0.677 ^d^	22.89 ± 0.024 ^e^	35.18 ± 1.636 ^cd^	43.84 ± 1.85 ^b^	41.60 ± 1.86 ^b^	35.78 ± 1.527 ^cd^	51.09 ± 1.786 ^a^
UFA, mg/g	78.61 ± 5.748 ^c^	73.58 ± 1.745 ^c^	47.58 ± 1.27 ^d^	71.45 ± 0.738 ^c^	88.39 ± 3.717 ^b^	87.42 ± 3.717 ^b^	75.3 ± 4.265 ^c^	110.39 ± 5.578 ^a^
TFA, mg/g	116.92 ± 7.913 ^c^	107.38 ± 2.423 ^c^	70.47 ± 1.294 ^d^	106.63 ± 2.374 ^c^	132.23 ± 5.574 ^b^	129.02 ± 5.574 ^b^	111.08 ± 5.793 ^c^	161.48 ± 7.365 ^a^

^1^ Each value is presented as mean ± standard deviation, ^a–f^ means in a row without a common superscript letter differ (*p* < 0.05), as analyzed by one-way ANOVA. ^2^ C14:0 = myristic acid, C16:0 = palmitic acid, C16:1 = palmitoleic acid, C17:0 = margaric acid, C18:0 = stearic acid, C18:1n9c = oleic acid, C18:2n6c = linoleic acid, C18:3n6 = methyl linolenate, C20:1 = eicosenoic acid, C18:3n3 = α-linolenic acid methyl ester, C20:2 = eicosadienoic acid, C20:3n6 = eicosatrienoic acid, C20:4n6 = methyl arachidonate, C22:6n3 = docosahexaenoic acid, SFA = saturated fatty acid, UFA = unsaturated fatty acid, TFA = total fatty acid. ^3^ LNV = low-dose normal vegetable oil, HNV = high-dose normal vegetable oil, LNL = low-dose normal lard, HNL = high-dose normal lard, LOV = low-dose oxidized vegetable oil, HOV = high-dose oxidized vegetable oil, LOL = low-dose oxidized lard, HOL = high-dose oxidized lard; different superscripts within a column indicate significant difference.

**Table 6 animals-11-00769-t006:** Effect of vegetable oil and lard with different qualities and concentrations on fatty acids of cooked yolk ^1^.

Item ^2^	Groups ^3^							
	LNV	HNV	LNL	HNL	LOV	HOV	LOL	HOL
C14:0, mg/g	0.44 ± 0.02 ^cd^	0.48 ± 0.03 ^bc^	0.4 ± 0.023 ^cd^	0.47 ± 0.05 ^bc^	0.54 ± 0.04 ^b^	0.84 ± 0.03 ^a^	0.47 ± 0.04 ^bc^	0.35 ± 0.03 ^d^
C16:0, mg/g	36.09 ± 1.76 ^ab^	37.18 ± 2.14 ^ab^	31.85 ± 2.54 ^b^	34.96 ± 3.33 ^ab^	39.08 ± 1.9 ^a^	36.14 ± 1.80 ^ab^	33.66 ± 2.96 ^ab^	26.5 ± 2.51 ^b^
C16:1, mg/g	3.86 ± 0.18 ^b^	4.31 ± 0.24 ^ab^	4.07 ± 0.31 ^b^	4.9 ± 0.46 ^a^	4.65 ± 0.23 ^ab^	3.99 ± 0.24 ^b^	4.71 ± 0.41 ^ab^	3.39 ± 0.32 ^b^
C17:0, mg/g	0.19 ± 0.01	0.22 ± 0.02	0.18 ± 0.02	0.17 ± 0.02	0.19 ± 0.01	0.51 ± 0.01	0.18 ± 0.02	0.18 ± 0.01
C18:0, mg/g	2.11 ± 0.13 ^c^	4.82 ± 0.03 ^a^	2.94 ± 0.07 ^bc^	2.54 ± 0.04 ^c^	2.95 ± 0.16 ^bc^	3.96 ± 0.17 ^b^	3.03 ± 0.09 ^b^	2.58 ± 0.05 ^c^
C18:1n9c, mg/g	54.32 ± 2.81	52.95 ± 3.61	46.48 ± 3.91	55.12 ± 5.48	55.6 ± 3.28	52.11 ± 2.38	46.87 ± 4.40	43.28 ± 4.32
C18:2n6c, mg/g	18.89 ± 0.90	23.56 ± 0.16	15.67 ± 1.24	14.54 ± 1.41	19.15 ± 1.07	20.21 ± 1.05	16.85 ± 1.52	12.69 ± 1.21
C18:3n6, mg/g	0.17 ± 0.01 ^c^	0.21 ± 0.02 ^b^	0.2 ± 0.02 ^bc^	0.16 ± 0.02 ^cd^	0.44 ± 0.02 ^a^	0.34 ± 0.01 ^a^	0.19 ± 0.01 ^bc^	0.13 ± 0.001 ^d^
C20:1, mg/g	0.32 ± 0.02	0.25 ± 0.01	0.23 ± 0.02	0.32 ± 0.04	0.29 ± 0.02	0.66 ± 0.01	0.28 ± 0.03	0.26 ± 0.04
C18:3n3, mg/g	0.41 ± 0.02 ^b^	0.67 ± 0.03 ^a^	0.32 ± 0.02 ^c^	0.3 ± 0.04 ^c^	0.44 ± 0.02 ^b^	0.30 ± 0.01 ^c^	0.4 ± 0.042 ^b^	0.25 ± 0.03 ^c^
C20:2, mg/g	0.20 ± 0.01 ^c^	0.24 ± 0.02 ^b^	0.15 ± 0.01 ^d^	0.17 ± 0.02 ^cd^	0.2 ± 0.01 ^c^	0.42 ± 0.01 ^a^	0.18 ± 0.02 ^cd^	0.15 ± 0.02 ^d^
C20:3n6, mg/g	0.25 ± 0.01	0.28 ± 0.02	0.25 ± 0.02	0.25 ± 0.02	0.28 ± 0.02	0.46 ± 0.01	0.27 ± 0.02	0.21 ± 0.01
C20:4n6, mg/g	0.64 ± 0.01	0.61 ± 0.05	0.61 ± 0.05	0.63 ± 0.09	0.62 ± 0.03	0.47 ± 0.01	0.65 ± 0.05	0.68 ± 0.06
C22:6n3, mg/g	1.21 ± 0.06 ^b^	1.62 ± 0.12 ^a^	0.89 ± 0.07 ^cd^	0.88 ± 0.09 ^cd^	1.15 ± 0.07 ^bc^	1.24 ± 0.06 ^b^	0.99 ± 0.09 ^c^	0.75 ± 0.01 ^d^
SFA, mg/g	39.02 ± 1.70 ^ab^	42.87 ± 2.46 ^a^	35.57 ± 2.54 ^ab^	38.24 ± 3.48 ^a^	42.96 ± 1.90 ^a^	41.45 ± 3.46 ^ab^	38.00 ± 3.2 ^ab^	29.78 ± 0.26 ^b^
UFA, mg/g	84.15 ± 4.25 ^ab^	88.76 ± 5.97 ^a^	72.23 ± 5.95 ^b^	80.67 ± 8.01 ^a^	86.6 ± 4.97 ^ab^	80.20 ± 3.46 ^ab^	74.76 ± 6.89 ^ab^	64.67 ± 6.26 ^b^
TFA, mg/g	123.17 ± 5.95 ^a^	131.63 ± 8.43 ^a^	107.8 ± 8.49 ^ab^	118.91 ± 11.49 ^a^	129.57 ± 6.88 ^a^	121.66 ± 7.89 ^a^	112.76 ± 10.10 ^ab^	94.45 ± 8.9 ^b^

^1^ Each value is presented as mean ± standard deviation, ^a–d^ means in a row without a common superscript letter differ (*p* < 0.05), as analyzed by one-way ANOVA. ^2^ C14:0 = myristic acid, C16:0 = palmitic acid, C16:1 = palmitoleic acid, C17:0 = margaric acid, C18:0 = stearic acid, C18:1n9c = oleic acid, C18:2n6c = linoleic acid, C18:3n6 = methyl linolenate, C20:1 = eicosenoic acid, C18:3n3 = α-linolenic acid methyl ester, C20:2 = eicosadienoic acid, C20:3n6 = eicosatrienoic acid, C20:4n6 = methyl arachidonate, C22:6n3 = docosahexaenoic acid, SFA = saturated fatty acid, UFA = unsaturated fatty acid, TFA = total fatty acid. ^3^ LNV = low-dose normal vegetable oil, HNV = high-dose normal vegetable oil, LNL = low-dose normal lard, HNL = high-dose normal lard, LOV = low-dose oxidized vegetable oil, HOV = high-dose oxidized vegetable oil, LOL = low-dose oxidized lard, HOL = high-dose oxidized lard; different superscripts within a column indicate significant difference.

## Data Availability

Data is contained within the article or Appendix A.

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
