# Peer review of "Effects of Lard and Vegetable Oils Supplementation Quality and Concentration on Laying Performance, Egg Quality and Liver Antioxidant Genes Expression in Hy-Line Brown"

_animals, 2021, doi:10.3390/ani11030769_

Round 1
Reviewer 1 Report
The work is interesting, the setting and methodology are correct; I suggest some aspects that can be improved:
1) the title could be more direct and short
2) the simple summary should be redrafted highlighting the aims, objectives and results.
3) the structure of the abstract should reflect the requirements of the journal:
- Background: Place the question addressed in a broad context and highlight the purpose of the study;
- Methods: Describe briefly the main methods or treatments applied. Include any relevant preregistration numbers, and species and strains of any animals used.
- Results: Summarize the article's main findings;
- Conclusion: Indicate the main conclusions or interpretations.
4) Discussion: a revision of the linguist form , some sentences are too long, would make it easier to read and understand
Author Response
The work is interesting, the setting and methodology are correct; I suggest some aspects that can be improved:
1) the title could be more direct and short
Answer: Done as requested. ‘Effects of lard and vegetable oils supplementation quality and concentration on laying performance, egg quality and liver antioxidant genes expression in Hy-line Brown
2) the simple summary should be redrafted highlighting the aims, objectives and results.
Answer: Done as requested. Simple summary: Adding oils into feeds is essential to growth and production performance of laying hens. As the main economic benefits of laying hens come from eggs, the quality assurance of eggs is crucial for producers. The term egg quality contains many indicators, including egg shape index, egg weight, yolk weight, yolk color, albumen height and haugh unit, which is an important index to measure the freshness of eggs. While the oils will oxidize during storage, and feeding with oxidized oil will affect the egg quality and nutritional value.. Herein, the Hy-line brown laying hens were fed diets with different types, concentrations and quality (normal or oxidized) of oil. The results showed that dietary oils quality significantly affect the egg qualities and the expression of liver antioxidant genes, providing useful information for laying hens.
3) the structure of the abstract should reflect the requirements of the journal:
- Background: Place the question addressed in a broad context and highlight the purpose of the study;
- Methods: Describe briefly the main methods or treatments applied. Include any relevant preregistration numbers, and species and strains of any animals used.
- Results: Summarize the article's main findings;
- Conclusion: Indicate the main conclusions or interpretations.
Answer: Thanks so much for the Reviewer’s comment and suggestion.
This study examined the effects of various types, quality and levels of dietary oils on laying performance and the expression patterns of antioxidant-related genes in Hy-line brown laying hens. Totally 720 40-week-old Hy-line brown laying hens were fed the same corn-soybean basal meals but containing 0.5% or 1.5% normal or oxidized soybean oil or lard, a total of 8 treatments. The results showed that laying rate (LR) and fatty acids of raw yolk were significantly correlated dietary type of oil (P < 0.05). With the increasing concentration of normal oil, it significantly in-creased LR and decreased feed conversion ratio (FCR, feed/egg) and albumen height of laying hens. The oxidized oil significant decreased the production performance of laying hens; and adding 1.5% of oxidized lard into feeds could destroy the integrity of yolk spheres of cooked yolk. mRNA expression of liver antioxidant-related genes increased when dietary oxidized oils were added into feeds. By comparing different qualities oil effect on antioxidant-related genes, the expression of Glutathione S-Transferase Theta 1(GSTT1), Glutathione S-Transferase Alpha 3 (GSTA3), Glutathione S-Transferase Omega 2 (GSTO2) and Superoxide Dismutase 2 (SOD2) were increased when dietary oils were oxidized, in which change of the GSTO2 expression was the most with 1.5% of oxidized soybean oil. In conclusion, the ideal type of oil for Hy-line brown layer hens is soybean comparing with lard in a corn-soybean diet, avoiding using of oxidized oil.
4) Discussion: a revision of the linguist form, some sentences are too long, would make it easier to read and understand
Answer: Done as requested. The discussion was partially revised.
The oxidized soybean oil and lard were murky and odorous liquid, indicating that insoluble substances and aldehydes were produced during the oxidation process [24]. The content of MDA that was as one of the final products of oil oxidation could illustrate the degree of oil oxidization [25]. The MDA content in oxidized lard increased by 6 times, however, the increase of oxidized soybean oil was smaller, possibly because the soybean oil contains antioxidant substances, which reduces the oxidation degree [26]. In addition, we could found that our treatments not isoenergetic or isoproteics from table 1, some effects could be the result of this in the subsequent analysis.
Comparing with all the normal oil groups, the ADFI and AEW both decreased in the oxidized oil groups. The explanation for this might be that oxidized oil decreased feed digestibility, damaged intestinal epithelial cells, affected feed absorption and usage, and de-creased production efficiency [27,28]. And the strong smell and taste of oxidized oil may also have contributed to the decrease the feed intake. In addition, with the same quality and concentration of oil, soybean oil as a feed additive can improve the laying rate of laying hens better than lard in our study. We suggested add soybean oil to the feed to improve the performance of laying hens, and the oil should be sealed and placed in a cool place to prevent oxidation. For the effect of different oil qualities on egg quality, the HU that reflects the protein content and freshness of the egg had increased significantly in HOV group than that in HNV group. The reason is that the ROS of oxidized oil could promotes the synthesis and secretion of antioxidant enzymes in the liver [29], and enhances the secretion capacity of the secretory cells in the magnum, which secreted the egg white to encase the yolk [30]. Therefore, when we evaluate the quality of eggs, we are not inclined to take HU as a single standard for evaluation.
Dietary oil could affect the lipid metabolism of laying hens to alter the lipid composition of yolk [4]. Rowghani et al. reported that a significant increase in cholesterol of yolk was observed in 24 weeks Hy laying hens when fed 3% or 5% level of canola oil [31]. The SFA content of raw yolk decreased and the UFA increased as the concentration of the same quality of oil increases in our study. And the fatty acid of raw yolk in the oxidized oil group is higher than in the normal oil group may be due to the lower LR, which increases the amount of fatty acid deposited. In addition, the α-linolenic acid (C18:3n3), eicosadienoic acid (C20:2) and docosahexaenoic acid (C22:6n3) of cooked yolk in soybean group is higher than in lard group. With the increasing concentration, the content of linoleic acid (C18:2n6c), palmitoleic acid (C16:1) and UFA had increased, which was consistent with the variation of fatty acid in raw yolk. The less content of fatty acid could was observed obviously in cooked yolk rather than in raw yolk, the reason is that yolk produced lots of volatile substances while the eggs were being cooked to reduce various of fatty acids [32]. Finally, the destruction of the integrity of yolk spheres were observed in the HOL group under the SEM.
Different dietary quality significantly affected antioxidant-related genes expression of livers in laying hens during the 3-weeks experimental period. From figure 1, we could found that the GSTT1, GSTA3, GSTO2 and SOD2 genes were significantly increased in different degrees as the oils oxidized. These data suggest a significantly positive feedback effect of the oxidation of oils on the antioxidant gene of liver. GSTT1, GSTA3 and GSTO2 genes belongs to glutathione transferase (GSTs) family [33], which are widely distributed in various cells and catalyze the endogenous and exogenous detrimental electrophiles to conjugate them with reduced glutathione [34], making the oxidized substances more easily pass through the cell membrane, thus being excluded from the body [35]. The discovery of SOD2, which specifically catalyze the dismutation of superoxide radicals (O2−) to hydrogen peroxide (H2O2) and oxygen. And the H2O2 can react with ferrous ion to form hydroxyl [36]. Moreover, the SOD2 gene has ability to remove the ROS that is produced by respiratory chain to maintain the stability of mitochondrial DNA [37].
Reviewer 2 Report
Dear Authors,
Please see my comments:
- Simple summary is well written. However, Adding one or two lines about the important obtained results could be more attractive for readers (optional).
- Introduction: It's not well organized. It looks it copied and paraphrased from somewhere else. Talk about the problem which is oxidative stress and say how oil supplementation could reduce the negative effects of oxidative stress. Moreover, in Line 64-67, This statement should go after line 50.
- I highly recommend you reorganize the introduction section and start with the problem then provide your solutions.
- Line 82, cage size
- Line 82. 'The entire chicken population' where these birds housed? pens or cages?
- Lines 88-91. Abbreviations are already explained under Table 1. Please remove from the text.
- The method and results sections are well written and explained.
Best regards,
Author Response
Simple summary is well written. However, Adding one or two lines about the important obtained results could be more attractive for readers (optional).
Answer: Done as requested.
Adding oils into feeds is essential to growth and production performance of laying hens. As the main economic benefits of laying hens come from eggs, the quality assurance of eggs is crucial for producers. The term egg quality contains many indicators, including egg shape index, egg weight, yolk weight, yolk color, albumen height and haugh unit, which is an important index to measure the freshness of eggs. While the oils will oxidize during storage, and feeding with oxi-dized oil will affect the egg quality and nutritional value.. Herein, the Hy-line brown laying hens were fed diets with different types, concentrations and quality (normal or oxidized) of oil. The results showed that dietary oils quality significantly affect the egg qualities and the expres-sion of liver antioxidant genes, providing useful information for laying hens.
Introduction: It's not well organized. It looks it copied and paraphrased from somewhere else. Talk about the problem which is oxidative stress and say how oil supplementation could reduce the negative effects of oxidative stress. Moreover, in Line 64-67, This statement should go after line 50. I highly recommend you reorganize the introduction section and start with the problem then provide your solutions.
Answer: Done as requested
Oil is an important supplementation for animal feed. Oil plays numerous roles, such as improving palatability, feed intake and the energy in feed, enhancing the immunity of body and decreasing the frequency of disease [1,2]. Subsequently, it was widely used in livestock and poultry feed and there is increasing concern the addition of oil in feed from poultry production. Currently, the effect of oil on production performance had been re-ported many times. The average daily gain (ADI) increased 10.8% and feed conversion ratio (FCR) decreased 11.5% with adding 2% peanut oil in the feed for broiler chickens [3]. Moreover, Adding 6% canola oil could significantly decrease the laying rate (LR), average egg weight (AEW) and average daily feed intake (ADFI) [4].
During the storage of oils, the unsaturated fatty acids (UFA) are easily affected by light, heat, oxygen, and microorganisms [5], causing oxidation reactions to generate per-oxide, malondialdehyde (MDA) and other oxides, which reduce the quality of oils and animal products [6]. The oxidation of oils in feed is one of the most common factors to cause the oxidative stress in animals [7]. Oxidative stress is a phenomenon caused by an imbalance between production and accumulation of reactive oxygen species (ROS) in cells and tissues and the ability of a biological system to detoxify these reactive products [8]. Therefore, the animal ingests oxidized oils, the peroxide, malondialdehyde (MDA) and other oxides enter digestive tract and are absorbed into blood to damage the cellular structure and affect various physiological functions [9]. In the process, the body creates mechanisms to respond to cellular oxidative stress, in which superoxide dismutase 2 (SOD2) converts the superoxide anion (O2-) in ROS to hydrogen peroxide (H2O2) [10]; un-der the action of catalase, H2O2 is oxidized into H2O and O2, which are finally utilized and cleared through water metabolism and respiration of the body [11]. Another way is to use glutathione-s-transferase (GST), thionredoxin, Vitamin E and other reducing substances to bind to ROS and block the chain reaction, so as to maintain cell stability [12]. Of these, Glutathione S-Transferase Theta 1(GSTT1), Glutathione S-Transferase Alpha 3 (GSTA3) and Glutathione S-Transferase Omega 2 (GSTO2) were distributed in cytoplasm, mitochondria, and cell membranes [13], covalently binding to glutathione to metabolize the intermediate products of oxidative stress [12]. Therefore, we speculated that the antioxidant gene expression level in laying hens significantly increased, indicating the occurrence of oxidative stress.
So far, oil that is the most commonly used energy feed is favored by producers [14], as it plays an important role in the production performance and egg quality of laying hens, and little work has been done to integrally assess the effect of oil on egg quality, production performance and expression of antioxidant genes of laying hens in details. The objective of the present study was to investigate the production performance and correlation of liver GSTT1, GSTA3, GSTO2 and SOD2 expression and explore the differences of egg quality with the lard and vegetable oil supplementation of different qualities and concentrations for Hy-line Brown laying hens.
Line 82, cage size
Answer: Done as requested. All the chickens housed in cages as say three per cage of 1350 cm2 separately under a light/dark cycle of 16 hours light and 8 hours dark (16L:8D), equipped with an individual feeder and water.
Line 82. 'The entire chicken population' where these birds housed? pens or cages?
Answer: Done as requested. All the chickens housed in cages as say three per cage of 1350 cm2 separately under a light/dark cycle of 16 hours light and 8 hours dark (16L:8D), equipped with an individual feeder and water.
Lines 88-91. Abbreviations are already explained under Table 1. Please remove from the text.
The method and results sections are well written and explained.
Answer: Done as requested. The abbreviations were removed from the text.
Reviewer 3 Report
animals-1101365
Effects of lard and vegetable oils supplementation quality and concentration on egg quality and liver antioxidant-related genes expression in Hy-line Brown
Originality/Novelty: The scientific question is original and well defined.
The results provide an advance in current knowledge.
Significance: Results are appropriately interpreted, and they are significant. The conclusions are justified and supported by the results
The hypothesis needs to be inserted in the introduction.
Quality of Presentation: Overall, the manuscript is written and well-organized. The manuscript presents some interesting data about the effects of lard and soybean oil in laying hens. The authors used 720 Hy-line brown laying hens distributed in 8 treatments and 3 replications. Assuming these numbers are correct each replication has 30 hens. It is unusual, but the authors did not insert extra information.
Scientific Soundness: the study is correctly designed; analyses are adequate but I wanted to see in this kind of research the oxidative assays of female blood or liver.
Interest to the Readers: the paper is attractive to animal production science readers and the poultry science community.
Overall Merit: the work provides an advance towards the current knowledge about the used oil, but the knowledge about the problems to use oxidized oil is very known.
English Level: The English language is appropriate and understandable.
Based on all the above I recommend this work for publication after revision.
Line 48-49- please insert some connective expression/word between sentences
Line 50-51 – authors are describing some results in poultry using oils in diet and change to oxidative stress without connections between paragraphs. Lines 53-54 make this connection, consider changing the order of sentences.
Lines 54-57 – and how about the aldehydes produced by oxidized oils?
Lines 68-69 – this sentence is here to authors justify their experiment; I do not believe that the authors decided to do this study because it is poorly studied. This is a weak reason, please consider thinking better and write again. What is a related-functional gene?
The authors studied just some genes that synthesize some of the enzymes from the antioxidant system, but either this paper is not in “details”.
What is the hypothesis tested here?
The conclusion needs to follow the hypothesis and answer it.
Material and methods
Line 81, insert 3 replicates with 10 hens.
Please insert the density of animal or the cage dimension and specification of the cage models (animal care)
About diet - insert conditions of storage, temperature, light protection, the period of storage, etc to avoid oxidization of “normal” diets.
Table 1 – please insert the antioxidant used,
explain if the nutrient composition were analyzed or calculated,
insert amino acids (at least the essentials) composition
describe the premix composition in microminerals and vitamins by kg of diet
Why authors insert in this table composition the value of linoleic acid? Is it the main in the lard too?
Treatment terminology: I don’t like the term “normal” because the other one needs to be “not normal”. I don’t have a suggestion to change here, but they are oxidized or not.
The same idea is to vegetable oil and lard, if one is vegetable other is animal, then here my suggestion is to use soybean oil.
The same is in all tables. I understand the option to use the abbreviation to describe the treatments, but it is really hard to follow. My recommendation is to use the abbreviation in text description but to cite Soybean oil and Lard over the 4 treatments in the first line, then in second-line just the expressions Normal (or the equivalent) and Oxidized over the 0,5 and 1,5% oil treatments; and in a third line just the oil % (0,5 and 1,5 %). In this organization, the first 4 columns will be to soybean oil and the next 4 to lard. But it is a suggestion.
In the discussion, authors need to mention that treatments were not isoenergetic or isoproteics, and some effects could be the result of this also, especially the treatments high oil with 16.83% of protein
Line 112 – correct the formula to HU, the egg weight is elevated to 0.37
Line 129 – Chinese
Line 145 – 3 replicate and only 3 females analyzed to mRNA?
Results
Table 3 – insert the unit g/g or kg/kg in FCR, or g/dozen?
I recommend to use the variable as the FI feed intake and cite the unit as g/day/hen
Figure 1 – the magnification is true only to files (if the SEM produces digital images) in the original size. In here probably the images were adjusted to the size of the plate. It is very important to include a scale bar in micrometers in each image or at least one for each magnification.
The authors made a lot of affirmation based on this image.
The same regions in images from HVO and HNV at 100x have the same “fragmentation” aspect that image from HOL at 300x magnification. In general, is hard to conclude that they are artifacts or unfocused areas. Please consider increasing the quality of images in pixels or the definition. It could be a result of the plate preparation.
Please insert in methodology how many SEM images were analyzed to define the size of spheres, and how many were measured per treatment.
Discussion
Line 300-304 – Oxidized oils have a strong smell and taste; did authors consider it as a direct fact to reduce the feed intake?
Lines 328-332 – based on this discussion I recommend in future to analyze the resistance of the vitelline membrane in fresh and stored eggs
Conclusion – there are a lot of results inside, please delete results and conclude, for example, the sentence about gene expression.
Author Response
Line 48-49- please insert some connective expression/word between sentences
Answer: Done as requested.
The average daily gain (ADI) increased 10.8% and feed conversion ratio (FCR) decreased 11.5% with adding 2% peanut oil in the feed for broiler chickens [3]. Moreover, Adding 6% canola oil could significantly decrease the laying rate (LR), average egg weight (AEW) and average daily feed intake (ADFI) [4].
Line 50-51 – authors are describing some results in poultry using oils in diet and change to oxidative stress without connections between paragraphs. Lines 53-54 make this connection, consider changing the order of sentences.
Answer: Done as requested.
We have re-described this paragraph.
During the storage of oils, the unsaturated fatty acids (UFA) are easily affected by light, heat, oxygen, and microorganisms [5], causing oxidation reactions to generate per-oxide, malondialdehyde (MDA) and other oxides, which reduce the quality of oils and animal products [6]. The oxidation of oils in feed is one of the most common factors to cause the oxidative stress in animals [7]. Oxidative stress is a phenomenon caused by an imbalance between production and accumulation of reactive oxygen species (ROS) in cells and tissues and the ability of a biological system to detoxify these reactive products [8]. Therefore, the animal ingests oxidized oils, the peroxide, malondialdehyde (MDA) and other oxides enter digestive tract and are absorbed into blood to damage the cellular structure and affect various physiological functions [9]. In the process, the body creates mechanisms to respond to cellular oxidative stress, in which superoxide dismutase 2 (SOD2) converts the superoxide anion (O2-) in ROS to hydrogen peroxide (H2O2) [10]; un-der the action of catalase, H2O2 is oxidized into H2O and O2, which are finally utilized and cleared through water metabolism and respiration of the body [11]. Another way is to use glutathione-s-transferase (GST), thionredoxin, Vitamin E and other reducing substances to bind to ROS and block the chain reaction, so as to maintain cell stability [12]. Of these, Glutathione S-Transferase Theta 1(GSTT1), Glutathione S-Transferase Alpha 3 (GSTA3) and Glutathione S-Transferase Omega 2 (GSTO2) were distributed in cytoplasm, mitochondria, and cell membranes [13], covalently binding to glutathione to metabolize the intermediate products of oxidative stress [12]. Therefore, we speculated that the antioxidant gene expression level in laying hens significantly increased, indicating the occurrence of oxidative stress.
Lines 54-57 – and how about the aldehydes produced by oxidized oils?
Answer: Done as requested.
During the storage of oils, the unsaturated fatty acids (UFA) are easily affected by light, heat, oxygen, and microorganisms [5], causing oxidation reactions to generate per-oxide, malondialdehyde (MDA) and other oxides, which reduce the quality of oils and animal products [6].
Lines 68-69 – this sentence is here to authors justify their experiment; I do not believe that the authors decided to do this study because it is poorly studied. This is a weak reason, please consider thinking better and write again. What is a related-functional gene?
Answer: We supplemented the reasons and renamed the related-functional gene to antioxidant genes.
‘So far, oil that is the most commonly used energy feed is favored by producers [13], as it plays an important role in the production performance and egg quality of laying hens, and little work has been done to integrally assess the effect of oil on egg quality, production performance and expression of antioxidant genes of laying hens in details.’
The authors studied just some genes that synthesize some of the enzymes from the antioxidant system, but either this paper is not in “details”.What is the hypothesis tested here? The conclusion needs to follow the hypothesis and answer it.
Answer: Thanks so much for the Reviewer’s comment and suggestion.
During the storage of oils, the unsaturated fatty acids (UFA) are easily affected by light, heat, oxygen, and microorganisms [5], causing oxidation reactions to generate per-oxide, malondialdehyde (MDA) and other oxides, which reduce the quality of oils and animal products [6]. The oxidation of oils in feed is one of the most common factors to cause the oxidative stress in animals [7]. Oxidative stress is a phenomenon caused by an imbalance between production and accumulation of reactive oxygen species (ROS) in cells and tissues and the ability of a biological system to detoxify these reactive products [8]. Therefore, the animal ingests oxidized oils, the peroxide, malondialdehyde (MDA) and other oxides enter digestive tract and are absorbed into blood to damage the cellular structure and affect various physiological functions [9]. In the process, the body creates mechanisms to respond to cellular oxidative stress, in which superoxide dismutase 2 (SOD2) converts the superoxide anion (O2-) in ROS to hydrogen peroxide (H2O2) [10]; un-der the action of catalase, H2O2 is oxidized into H2O and O2, which are finally utilized and cleared through water metabolism and respiration of the body [11]. Another way is to use glutathione-s-transferase (GST), thionredoxin, Vitamin E and other reducing substances to bind to ROS and block the chain reaction, so as to maintain cell stability [12]. Of these, Glutathione S-Transferase Theta 1(GSTT1), Glutathione S-Transferase Alpha 3 (GSTA3) and Glutathione S-Transferase Omega 2 (GSTO2) were distributed in cytoplasm, mitochondria, and cell membranes [13], covalently binding to glutathione to metabolize the intermediate products of oxidative stress [12]. Therefore, we speculated that the antioxidant gene expression level in laying hens significantly increased, indicating the occurrence of oxidative stress.
Material and methods
Line 81, insert 3 replicates with 10 hens.
Please insert the density of animal or the cage dimension and specification of the cage models (animal care)
About diet - insert conditions of storage, temperature, light protection, the period of storage, etc to avoid oxidization of “normal” diets.
Answer: Thanks so much for the Reviewer’s comment and suggestion.
A total of 720 Hy-line brown layers of 40 weeks based upon similar body weight of 2000.79 ± 166.18 g/bird were selected from breed base of China Agriculture University (Zhuozhou, China) and randomly assigned to eight dietary treatments, three replicates per treatment, 3 replicates with 90 hens. All the chickens housed in cages as say three per cage of 1350 cm2 separately under a light/dark cycle of 16 hours light and 8 hours dark (16L:8D), equipped with an individual feeder and water. Birds were allowed to adapt to experimental diets in the layer house for 1 week; the entire experiment lasted for 3 weeks. According to the assigned treatment groups, layers were fed with corn-soybean meals containing either 0.5 or 1.5% of normal or oxidized lard or soybean oil, the corn-soybean feed was stored in a dark, 4 °C, and was produced within one month place.
Table 1 – please insert the antioxidant used, explain if the nutrient composition were analyzed or calculated, insert amino acids (at least the essentials) composition describe the premix composition in microminerals and vitamins by kg of diet
Answer: Thanks so much for the Reviewer’s comment and suggestion.
Premix: Mineral premix provided per kg of diet: Mn, 100 mg; Fe, 80 mg; Zn, 75 mg; Cu, 8 mg; I, 0.35 mg; Se, 0.15 mg. Vitamin premix provided per kg of diet: vitamin A, 12,500 IU; vitamin D3, 2,500 IU; vitamin E, 30 IU; vitamin K3, 2.65 mg; vitamin B1, 2 mg; vitamin B2, 6 mg; vitamin B12, 0.025 mg; biotin, 0.0325 mg; folic acid, 1.25 mg; pantothenic acid, 12 mg; niacin, 50 mg.
Antioxidant = ethoxyquin
Why authors insert in this table composition the value of linoleic acid? Is it the main in the lard too?
Answer: Thanks so much for the Reviewer’s comment and suggestion. We had deleted the value of linoleic acid.
Treatment terminology: I don’t like the term “normal” because the other one needs to be “not normal”. I don’t have a suggestion to change here, but they are oxidized or not. The same idea is to vegetable oil and lard, if one is vegetable other is animal, then here my suggestion is to use soybean oil. The same is in all tables. I understand the option to use the abbreviation to describe the treatments, but it is really hard to follow. My recommendation is to use the abbreviation in text description but to cite Soybean oil and Lard over the 4 treatments in the first line, then in second-line just the expressions Normal (or the equivalent) and Oxidized over the 0,5 and 1,5% oil treatments; and in a third line just the oil % (0,5 and 1,5 %). In this organization, the first 4 columns will be to soybean oil and the next 4 to lard. But it is a suggestion.
Answer: Thanks so much for the Reviewer’s comment and suggestion.
In the discussion, authors need to mention that treatments were not isoenergetic or isoproteics, and some effects could be the result of this also, especially the treatments high oil with 16.83% of protein
Answer: Thanks so much for the Reviewer’s comment and suggestion. We had supplemented this description.
The oxidized soybean oil and lard were murky and odorous liquid, indicating that insoluble substances and aldehydes were produced during the oxidation process [24]. The content of MDA that was as one of the final products of oil oxidation could illustrate the degree of oil oxidization [25]. The MDA content in oxidized lard increased by 6 times, however, the increase of oxidized soybean oil was smaller, possibly because the soybean oil contains antioxidant substances, which reduces the oxidation degree [26]. In addition, we could found that our treatments not isoenergetic or isoproteics from table 1, some effects could be the result of this in the subsequent analysis.
Line 112 – correct the formula to HU, the egg weight is elevated to 0.37
Answer: Done as requested.
Line 129 – Chinese
Answer: Done as requested.
Line 145 – 3 replicate and only 3 females analyzed to mRNA?
Answer: At the end of the trial, we randomly selected a chicken from each replicate and killed the chickens by cervical dislocation, a total of 24 chickens
Results
Table 3 – insert the unit g/g or kg/kg in FCR, or g/dozen?
I recommend to use the variable as the FI feed intake and cite the unit as g/day/hen
Answer: Done as requested. FCR, g/g
Figure 1 – the magnification is true only to files (if the SEM produces digital images) in the original size. In here probably the images were adjusted to the size of the plate. It is very important to include a scale bar in micrometers in each image or at least one for each magnification. The authors made a lot of affirmation based on this image. The same regions in images from HVO and HNV at 100x have the same “fragmentation” aspect that image from HOL at 300x magnification. In general, is hard to conclude that they are artifacts or unfocused areas. Please consider increasing the quality of images in pixels or the definition. It could be a result of the plate preparation. Please insert in methodology how many SEM images were analyzed to define the size of spheres, and how many were measured per treatment.
Answer: Done as requested.
We re-produced the figure 1. First, we added the a scale bar in micrometers for each magnification. We also change the HNV at 100x and HOL at 300x magnification. The figure of HOV at 100x magnification is the clearest. Finally, we supplemented statement at Materials and Methods ‘A Hitachi SU8010 SEM (Hitachi High‐Technologies, Tokyo, Japan) was used in this study with instructions. For each sample observed by SEM, micrographs at different mag-nifications (100x, 300x, 700x) were recorded. Each magnification that had 3 micrographs was analyzed, indicating each treatment with 9 micrographs.’ Base on this, we defined the size of yolk spheres.
Discussion
Line 300-304 – Oxidized oils have a strong smell and taste; did authors consider it as a direct fact to reduce the feed intake?
Answer: Thanks so much for the Reviewer’s comment and suggestion.
Comparing with all the normal oil groups, the ADFI and AEW both decreased in the oxidized oil groups. The explanation for this might be that oxidized oil decreased feed di-gestibility, damaged intestinal epithelial cells, affected feed absorption and usage, and de-creased production efficiency [27,28]. And the strong smell and taste of oxidized oil may also have contributed to the decrease the feed intake.
Lines 328-332 – based on this discussion I recommend in future to analyze the resistance of the vitelline membrane in fresh and stored eggs
Answer: Thanks so much for the Reviewer’s comment and suggestion. We deleted these descriptions.
Conclusion – there are a lot of results inside, please delete results and conclude, for example, the sentence about gene expression.
Answer: Thanks so much for the Reviewer’s comment and suggestion. We re-described the conclusion.
’ In conclusion, under the same quality and concentration of oil, adding soybean oil to the feed of Hy-brown laying hens can improve production performance better than lard. Moreover, the expression level of antioxidant genes was significantly increased after feed-ing with oxidized oil, we should avoid utilizing oxidized oil to add into the feed to prevent the oxidative stress in vivo. Our findings provided new insights into the effect of different types, concentrations and qualities of dietary oils on the production performance and egg quality in laying hens.’